# Association between non-invasive biomarkers and quality of life in Primary Sclerosing Cholangitis

Jingyu Dai[1,2,3☯], Emmanuel Selvaraj[4☯], Emma L. Culver[2,5], Adam Bailey[2,5], Michael Brady[6], Michael Pavlides[2,4,7‡], Jose Leal[1,2‡*]

1 Department of Primary Care Health Sciences, University of Oxford, Oxford, United Kingdom, 2 Oxford National Institute for Health and Care Research (NIHR) Biomedical Research Centre, University of Oxford, Oxford, United Kingdom, 3 Oxford Health National Institute for Health and Care Research (NIHR) Biomedical Research Centre, University of Oxford, Oxford, United Kingdom, 4 South Warwickshire University Hospital NHS Foundation Trust, Warwick, United Kingdom, 5 Translational Gastroenterology and Liver Unit, University of Oxford, Oxford, United Kingdom, 6 Perspectum Ltd, Oxford, United Kingdom, 7 Radcliffe Department of Medicine, University of Oxford, Oxford, United Kingdom

☯ These authors contributed equally to this work.
‡ MP and JL also contributed equally to this work.
* jose.leal@phc.ox.ac.uk

## Abstract

### Background & aims

Primary sclerosing cholangitis (PSC) is a rare chronic liver disease that impact quality of life (QoL). This study investigated the association between biomarkers of PSC disease severity and QoL.

### Methods

Prospective study involving 80 participants with PSC at baseline and 55 at 1-year follow-up. QoL was assessed using patient reported outcomes (PROMs): RAND-SF36, SF6D and PSC-PRO. MRI-MRCP data was analysed using LiverMultiscan for iron-corrected liver T1 (cT1) and MRCP+ for the relative severity of intrahepatic biliary dilatations (RSIBD). Disease severity was also classified using FibroScan liver stiffness (LS), enhanced liver fibrosis (ELF), Mayo risk score (MRS), Amsterdam-Oxford model (AOM), alkaline phosphatase (ALP), presence of extrahepatic disease and dominant stricture (DS). Descriptive and regression analyses were conducted.

### Results

At baseline, more advanced PSC was associated with differences in PROMs, AOM>2, (PSC-PRO PSC symptoms, 5.181, p=0.048), LS>9.6 kPa (SF-6D, −0.081, p=0.027), cT1>825ms (SF6D QoL, −0.161, p=0.004; SF36 PCS, −10.595, p=0.001; SF36 MCS, −10.726, p=0.012), DS (PSC-PRO symptoms scores, 5.800,

**Data availability statement:** All relevant data underlying the findings of this study are available from the Open Science Framework (OSF) database (DOI: https://doi.org/10.17605/OSF.IO/BH94N ).

**Funding:** JD, JL, ELC, MP and AB were supported by the Oxford National Institute for Health and Care Research (NIHR) Biomedical Research Centre. ELC receives funding from PSC Support and the Oxford Health Service Research Committee. The funders had no role in study design, data collection and analysis, decision to publish, or preparation of the manuscript.

**Competing interests:** I have read the journal's policy and the authors of this manuscript have the following competing interests: MB is employed by Perspectum Ltd and is a shareholder in Perspectum Ltd, MP is a shareholder in Perspectum Ltd. ELC: Speaking Fees: Advanz (Intercept), Albireo, Dr Falk Pharma, Gilead, GSK, Mirum; Consulting Fees: Advanz (Intercept), Amgen (Horizon Therapeutics), Ipsen, Mirum, Moderna, Sanofi, Zenus Pharma, Sail; Grant Support: Jansen, Innovate UK, PSC Support; Institutional Funding Support: BRC Oxford NIHR (UK), Oxford Health Service Research Committee Grant. This does not alter our adherence to PLOS ONE policies on sharing data and materials." "JD, ES and JL report no conflict of interest.

**Abbreviations:** AOM, Amsterdam-Oxford model; ALP, Alkaline phosphatase; ELF, Enhanced liver fibrosis; ERCP, Endoscopic retrograde cholangiopancreatography; LS, FibroScan liver stiffness; MASLD, Metabolic dysfunction associated steatotic liver disease; MCS, Mental component summary; MCAR, Missing completely at random; MI, Multiple imputation; MRS, Mayo risk score; MRCP, Magnetic resonance cholangiopancreatography; NAFLD, Non-alcoholic fatty liver disease; PCS, Physical component summary; PROMs, Patient-reported outcome measures; PSC, Primary sclerosing cholangitis; QoL, Quality of Life; ULN, Upper limit of normal.

p = 0.025), RSIBD (SF-6D, −0.081, p = 0.016). During follow-up, increase in LS was associated with a reduction in QoL measured via SF-6D (−0.002, p < 0.001), SF36 physical component summary (−0.246, p < 0.001) and SF36 mental component summary (−0.171, p < 0.001).

## Conclusions

QoL in PSC was associated with biomarkers of parenchymal liver fibrosis (LS, cT1), biliary disease (dominant strictures, RSIBD), and composite scores of disease severity (AOM). Increasing LS predicted further declines in QoL. Further research should explore MRCP+ metrics and their impact on QoL.

## 1. Introduction

Primary Sclerosing cholangitis (PSC) is a rare chronic liver disease that is characterized by intrahepatic and extrahepatic bile duct inflammation and fibrosis leading to multifocal biliary strictures, progressive cholestasis and an increased risk of malignancy [1,2]. PSC is closely associated with inflammatory bowel disease, with ulcerative colitis the predominant phenotype in 70–80%. PSC is characterised by the occurrence of clinical events such as biliary obstruction, cholangitis, liver cirrhosis with decompensation, and both hepatobiliary and colorectal malignancy [2]. Liver transplantation is the only life-extending option in those with advanced disease [3]. Median time to liver transplantation ranged from 9.7 to 20.6 years and there was a 30–50% risk of PSC recurrence following transplantation [3,4].

Determining prognosis in PSC remains is an important clinical question. Several methods for determining risk have been described, including multi-variate prognostic models and other non-invasive biomarkers. Two of the most validated prognostic models are the revised Mayo Risk score [MRS] [5] and Amsterdam-Oxford model [AOM] [6] while non-invasive biomarkers such as serum alkaline phosphatase (ALP), liver stiffness measured using FibroScan (LS), and the enhanced liver fibrosis (ELF) have also been shown to carry prognostic information [1,7]. Furthermore, magnetic resonance imaging with magnetic resonance cholangiopancreatography (MRI-MRCP) is the preferred way of diagnosing, describing features of cholangiopathy and assessing biliary complications. The recent development of MRCP+ metrics offers a more objective and nuanced assessment of a biliary tree measurement and have been shown to correlate with other markers of disease severity [1]. Such metrics have the potential to improve the risk-stratification of people with PSC.

People with PSC report lower quality of life than healthy controls and similar quality of life to those with other chronic conditions [8]. However, there is considerable uncertainty on how staging and prognosis methods and biomarkers correlate with the lived experiences and quality of life of people with PSC. The aim of our study was to: (1) investigate whether non-invasive biomarkers facilitate stratifying participants with PSC according to their quality of life; (2) identify the association between changes in biomarkers and changes in the quality of life of participants with PSC.

## 2. Methods

### 2.1 Study sample

This was a prospective, single-centre, longitudinal study at Oxford University Hospital (OUH) NHS Foundation Trust. Recruitment started on 6th Sept 2018 and is ongoing, our study is presenting an adhoc interim analysis of QoL data. Participants gave written informed consent. Adult participants (aged 18 years or older) with known diagnosis of large-duct PSC were recruited from outpatient clinics over two years (September 2018 to November 2020). Large-duct PSC was defined as the association of chronic cholestasis with typical features on magnetic resonance cholangiopancreatography (MRCP) or endoscopic retrograde cholangiopancreatography (ERCP), and with no cause of secondary sclerosing cholangitis.

Clinical, laboratory, imaging and QoL data were collected both at baseline and 1 year follow-up. Qualitative MRI-MRCP reads were performed only at baseline. Quantitative MRI-MRCP (LiverMultiscan and MRCP+) analysis were performed at both time points. Data collected are summarised in S1 Table.

The study design and protocol have been described in detail previously [1]. The UK Research Ethics service (18/SC/0367) and local Research and Development approved this study. All participants gave written informed consent.

### 2.2 Risk stratification

Published criteria were used to stratify participants with PSC into groups at high and low risk of progression according to: prognostic risk models (MRS and AOM); serum ALP; liver fibrosis markers; disease distribution (extrahepatic biliary disease) and dominant stricture, and MRCP+ metrics [1].

High-risk groups were defined as follows: MRS > 0 [9,10], AOM > 2 [11], LS > 9.6kPa [12], ELF > 9.8 [13], ALP > 1.5x ULN [14], ALP > 2.2x ULN [15], presence of extrahepatic disease [15], and presence of dominant stricture [16] (see Table 1).

### 2.3 Patients reported outcome measures (PROMs)

**2.3.1 PSC-specific patient reported outcome (PSC-PRO).** The PSC-PRO is a patient-reported outcome instrument specific for people with PSC. PSC-PRO reflects the experience of people with PSC on disease and treatments, both

**Table 1. Definition of low and high-risk groups in PSC.**

| Study variable | Detail | High-risk threshold |
|---|---|---|
| Clinical data | IBD† | Present |
| Cholestatic marker | Serum ALP‡ | > 1.5x ULN ¶<br>> 2.2x ULN ¶ |
| Prognostic model | MRS | > 0 |
| | AOM § | > 2 |
| Liver fibrosis marker | LS | > 9.6kPa |
| | ELF * | > 9.8 |
| Qualitative MRI-MRCP | Anali score | > 2 |
| | Extrahepatic disease | Present |
| | Dominant stricture | Present |
| | Cirrhosis | Present |
| LiverMultiscan | Median iron-corrected liver T1 (cT1) | cT1 > 825ms |
| MRCP+ | Relative severity of intrahepatic biliary dilatations (RSIBD) | RSIBD > 7 |

†IBD, inflammatory bowel disease; ‡ALP; alkaline phosphatase; §AOM, Amsterdam-Oxford model; * ELF, enhanced liver fibrosis; ¶ ULN, upper limit of normal.

physically and mentally. It consists of two modules: 1) PSC symptoms within the last 24 hours, 2) impact of symptoms in the last 7 days. Module 1 comprises 13 questions concerning variety of PSC symptoms severity from 0 (no symptoms) to 10 (symptoms as bad as you could imagine). Module 2 consists of seven domains: physical function, activities of daily living, work productivity, role function, emotional impact, social/leisure impact, and quality of life. Each question in each domain is scored from 1 ("Never") to 5 ("Always") in terms of the impact of symptoms. Module 1 score is estimated by adding up each item score for that module and ranges from 0 (no symptoms) to 120. Module 2 score is estimated by averaging item scores in each domain and summing the domain means to obtain an overall Impact of Symptoms score [17]. Higher scores indicate poorer QoL.

**2.3.2 RAND SF-36 and SF-6D.** The RAND 36-Item Short Form Survey Instrument (SF-36, RAND 36-Item Health Survey 1.0 Questionnaire) is a 36-item generic health measurement that contains eight scales: physical functioning (PF), role physical (RF), bodily pain (BP), general health (GH), vitality (V), social functioning (SF), role emotional (RE) and mental health (MH). The recall period for the SF-36 is 4 weeks. Each item is scored 100, 75, 60, 25 and 0, where 0 is the worst and 100 is the best health status. Physical component summary (PCS) and mental component summary (MCS) scores are aggregated by applying a scoring algorithm combines weighted contributions from all eight health scales. This allowed focusing the analysis on two outcomes rather than eight (for each scale) and avoid the decrease in power from multiple testing. To obtain PCS and MCS scores, we used the US population mean and standard deviation, as suggested by [18], and the respective algorithm [18,19]. RAND SF-36 lower scores indicate poorer QoL.

We also converted the RAND SF-36 answers into Short Form 6 Dimension (SF-6D) scores [20] using an algorithm provided by the University of Sheffield. The SF-6D score is a preference-based measure of health needed to estimate quality adjusted life years (QALYs) and inform on the cost-effectiveness of healthcare interventions. With the SF-6D, a score of 1 corresponds to perfect health and a score of 0 to death. Hence, lower SF-6D scores indicate poorer QoL.

## 2.4 Statistical analysis

Descriptive statistics of the recruited sample were used to summarise patient characteristics and biomarkers at the baseline visit. We assessed whether there were baseline differences in characteristics, risk factors and QoL between the recruited sample and the participants who attended both baseline and year 1 visits and provided PROM data.

Differences in quality of life (QoL) outcomes (PSC-PRO, SF-36, and SF-6D) between baseline and year 1 visits were estimated using ordinary least squares (OLS) regression. We also examined whether PSC disease severity at baseline, as defined by biomarkers, was associated with QoL reported at baseline and year 1 visits, separately. Associations between changes in QoL from baseline to year 1 visit and biomarker values at baseline were also estimated via OLS. Variation in QoL within participants, was assessed visually by plotting their quality of life at baseline and year 1 visits.

We assessed QoL jointly at baseline and year 1 visits by estimating the association between changes in disease progression biomarkers and quality of life outcomes (SF-6D, SF-36 PCS and MCS, and PSC-PRO symptoms and total impact of symptoms). S1 Table reports the biomarkers used. The association was estimated using a fixed effects (FE) model with robust standard errors. The FE model has a significant advantage over other regression methods (such as pooled OLS) given its ability to reduce the risk of omitted variable bias [21]. This approach uses each participant as their own control when estimating the impact of changes on the covariates of interest on the quality of life of the participants. As sensitivity analysis, we also included the eight scale scores from SF-36 and the seven domains from module 2 of the PSC-PRO. We also conducted univariate analysis to identify which predictors were significantly associated with changes in health outcomes between baseline year and year 1. Then, we performed multivariable analyses by including variables found to be significant in the univariate analysis into the multivariable models. We assessed the suitability of the FE model by conducting the Hausman test [22] and comparing the model coefficients with those of a random effects and pooled OLS models.

The small size of the analysis sample required additional caution when interpreting results from the FE models. Hence, in our study, the final predictors had to be significant in more than 50% of re-estimated models across 1000 bootstrap samples and remain significant after removing potential outliers (see S1 and S2 Text for details of the approach).

Finally, we also assessed the missing data mechanism by estimating the likelihood of not attending year 1 visit conditional on baseline risk factors and QoL (using a logistic regression). Data were not missing completely at random (MCAR) if the association was significant ($p < 0.05$). Conditional on data not being MCAR, multiple imputation (MI) was used as sensitivity analysis (see S3 Text and S2 Table for more details). Models were re-estimated by using the imputed dataset.

All analyses were performed using STATA 18MP. Differences were judged to be statistically significant if *p-value* was less than 0.05.

## 3. Results

### 3.1 Descriptive statistics

Eighty participants were recruited and had baseline assessments and 55 (69%) attended the Year 1 visit. PROMs were collected for 75 participants at baseline and 51 participants at the year 1 visit (S1 Fig). The reason for the low follow-up rate is that the follow-up period coincided with the COVID-19 pandemic, which meant that due to lockdown measures that were in place at various points in the study, some participants opted not to attend their follow-up visits.

The average age at baseline was 43 years old, with mean of 8 years of PSC duration, 90% were white and 68% were male (Table 2). We found no significant differences of biomarkers values and participants' characteristics at baseline between the recruited sample (n = 80) and those who attended both baseline and year 1 visits while providing PROMs data (n = 51, see S3 Table).

### 3.2 Self-reported quality of life

At baseline, the PROM scores did not differ significantly between the recruited sample with baseline PROMs (n = 75) and the analysis sample (n = 51, see S4 Table). Table 3 shows a decrease in quality of life between baseline and year 1 visits for the majority of PROMS. The exception was for PSC-PRO PSC symptoms, which showed an improvement between the two visits. However, none of the changes in PROMs scores were statistically significant between visits (SF-6D, 0.01, p = 0.733; SF-36 PCS, 1.23, p = 0.480; SF-36 MCS, 0.64, p = 0.777; PSC-PRO PSC Symptoms, 0.11, p = 0.960; PSC-PRO: Total Impact of Symptoms, −0.41, p = 0.573.

**Table 2. Descriptive statistics at baseline.**

|  | Recruited sample | Recruited sample with PROMs at baseline | Recruited sample with PROMs at baseline and year 1 (analysis sample) |
|---|---|---|---|
| N | 80 | 75 | 51 |
| White, n(%) | 72 (90.0%) | 68 (90.7%) | 47 (92.2%) |
| Male, n(%) | 54 (67.5%) | 51 (68.0%) | 34 (66.7%) |
| IBD presence, n(%) | 61 (76.2%) | 57 (76.0%) | 34 (66.7%) |
| Age, mean(SD) | 43.3 (16.7) | 42.5 (16.2) | 44.9 (15.8) |
| PSC duration, mean(SD) | 8.5 (5.9) | 8.3 (5.7) | 9.0 (6.3) |
| LSM, mean(SD) | 10.3 (8.5) | 10.5 (8.7) | 9.5 (6.9) |
| ELFS, mean(SD) | 9.6 (1.2) | 9.6 (1.1) | 9.5 (1.0) |
| xULN ALP, mean(SD) | 1.5 (1.3) | 1.6 (1.4) | 1.3 (0.9) |
| AOM, mean(SD) | 1.7 (0.7) | 1.7 (0.7) | 1.7 (0.6) |
| MRS, mean(SD) | 0.1 (0.8) | 0.1 (0.8) | 0.1 (0.6) |

**Table 3. Health outcomes at baseline and year 1 visits †.**

| Variable | Baseline | Baseline | Year 1 | Mean Difference between baseline and year 1 (p-value) |
|---|---|---|---|---|
| | Recruited sample with PROMs at baseline | Recruited sample with PROMs at baseline and year 1 (analysis sample) | Recruited sample with PROMs at baseline and year 1 (analysis sample) | Recruited sample with PROMs at baseline and year 1 (analysis sample) |
| N | 75 | 51 | 51 | 51 |
| SF6D Quality of life | 0.81(0.15) † | 0.81 (0.14) | 0.80 (0.16) | 0.01 (p = 0.73) |
| SF36: Physical component summary | 49.66 (8.29) | 50.02 (8.06) | 48.79 (9.83) | 1.23 (p = 0.48) |
| SF36: Mental component summary | 49.11 (10.55) | 49.03 (11.30) | 48.39 (11.98) | 0.64 (p = 0.78) |
| PSC-PRO: PSC Symptoms | 1.97 (10.55) | 2.85 (12.69) | 2.74 (7.81) | 0.11 (p = 0.96) |
| PSC-PRO: Total Impact of Symptoms | 8.96 (3.38) | 9.24 (3.46) | 9.65 (3.91) | −0.41 (p = 0.57) |

† *All values are showing in Mean (SD) unless described otherwise.*

Fig 1 shows the traces of changes in quality of life (SF-36, SF-6D and PSC-PRO) between baseline and year 1 visits (n = 51). We found considerable variation within participants' scores across all PROMs (see S5 Table for more detail).

### 3.3 Associations of PSC disease severity at baseline with QoL at baseline

Participants with AOM > 2 (high risk of disease progression) reported higher PSC-PRO PSC symptoms score compared to those at low risk (5.181, p = 0.048). When disease severity was assessed using transient elastograhy, participants with more advanced disease (LS > 9.6) reported lower SF-6D (−0.081, p = 0.027) and SF-36 PCS (−4.996, p = 0.018) compared to those with less advanced disease (LS ≤ 9.6). Participants with dominant stricture reported higher PSC-PRO symptoms scores (5.800, p = 0.025). Participants with more advanced intrahepatic biliary disease (RSIBD score >7) reported lower SF-6D scores (−0.081, p = 0.016). Finally, participants with cT1 > 825ms, reported lower SF6D Qol (−0.161, p = 0.004), lower SF36 PCS (−10.595, p = 0.001), lower SF36 MCS (−10.726, p = 0.012). Fig 2 provides the mean differences (95% CI) in PROMs scores at baseline between the disease severity groups. S6 Table reports the baseline scores for each PROM and risk group.

### 3.4 Associations between disease severity at baseline and QoL after 1 year

Participants with ALP > 2.2x ULN at baseline reported higher PSC-PRO PSC symptoms scores (6.735, p = 0.046) after 1 year. For MRCP features, participants with extrahepatic disease at baseline (high risk group) reported higher PSC-PRO PSC symptoms scores (5.269, p = 0.016) and lower SF36 PCS (−5.750, p = 0.038) after 1 year. Participants with LS > 9.6 reported lower SF36 PCS (−6.536, p = 0.030), lower SF36 MCS (−7.800, p = 0.034) and higher PSC-PRO total impact of symptom (2.915, p = 0.014) after 1 year. Participants with ELF > 9.8 at baseline reported lower PSC-PRO total impact of symptom (3.018, p = 0.011) after 1 year. Participants with dominant stricture at baseline reported higher PSC-PRO PSC symptom scores (4.583, p = 0.048) and total impact of symptoms (2.850, p = 0.013), and lower SF-36 PCS (−8.394, p = 0.003) and SF-6D score (−0.101, p = 0.029) after 1 year. Finally, participants with cT1 > 825ms, reported lower SF6D QoL (−0.190, p = 0.004), lower SF36 PCS (−15.489, p < 0.001), higher PSC-PRO PSC symptom scores (11.659, p < 0.001) and total impact of symptoms (3.682, p = 0.029) after 1 year. S7 Table reports PROM scores at year 1 visit by baseline risk groups.

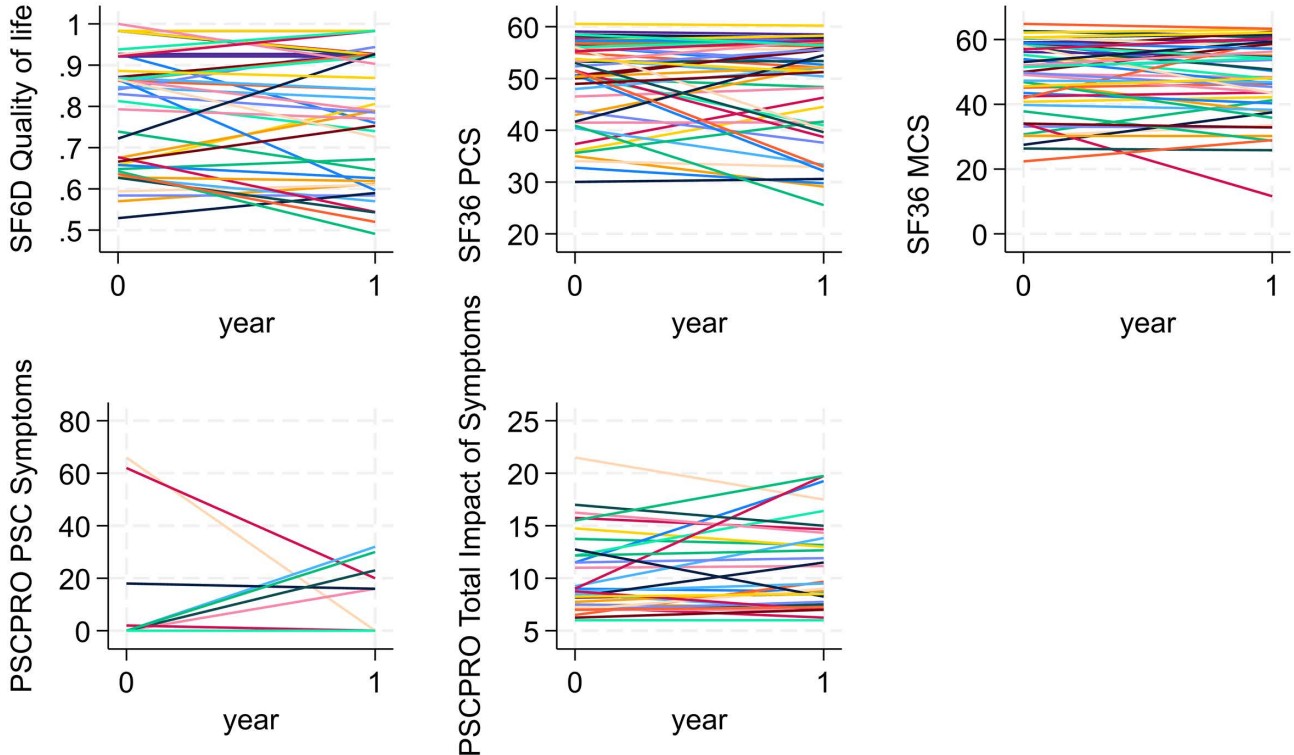

**Fig 1. Traces of participants' change in patient-reported outcomes between baseline and Year 1.** SF36 PCS: Physical component summary; SF36 MCS: SF 36 Mental component summary.

### 3.5 Associations between PSC disease severity at baseline and changes in QoL between baseline and 1-year

Participants with ALP ≤ 2.2 ULN reported a decrease in SF6D QoL (worse symptoms) while participants with ALP > 2.2 ULN reported an increase in score (better symptoms) with the difference between the two groups being statistically significant (0.081, p = 0.048). Furthermore, those with ELF ≤ 9.8 reported an increase (2.098, p = 0.008) in PSC-PRO total impact of symptoms (worse quality of life). Participants with IBD reported increase in SF36 MCS compare with participants without IBD (4.091, p = 0.036). Participants with cT1 > 825ms reported decrease in SF36 PCS (−6.762, p = 0.021), and increase in PSC-PRO symptoms (13.651, p = 0.034) and PSC-PRO total impact of Symptoms (3.854, p < 0.001) compared with cT1 <=825ms. S8 Table reports the change in PROMs scores between baseline and year 1 visits stratified by disease severity at baseline.

### 3.6 Association of changes in biomarkers with changes in quality of life

Changes to Fibroscan liver stiffness (LS) were found to be significantly associated with changes in participants' PROMs scores and passed the two robustness checks (i.e., significant in over 50% of bootstrap samples and significant after removing 5 outliers, see S4 Text for more details). The Hausman test suggested the random effects estimator to be more efficient in the univariate (p > 0.05) and multivariate analyses (p > 0.05).

In univariate analysis, we found that as LS increased the participants' QoL decreased in terms of SF-36 PCS (−0.246, p < 0.001), SF-36 MCS (−0.243, p < 0.001), and SF-6D (−0.002, p < 0.001). We did not find a significant association between LS and the two PSC-PRO outcomes.

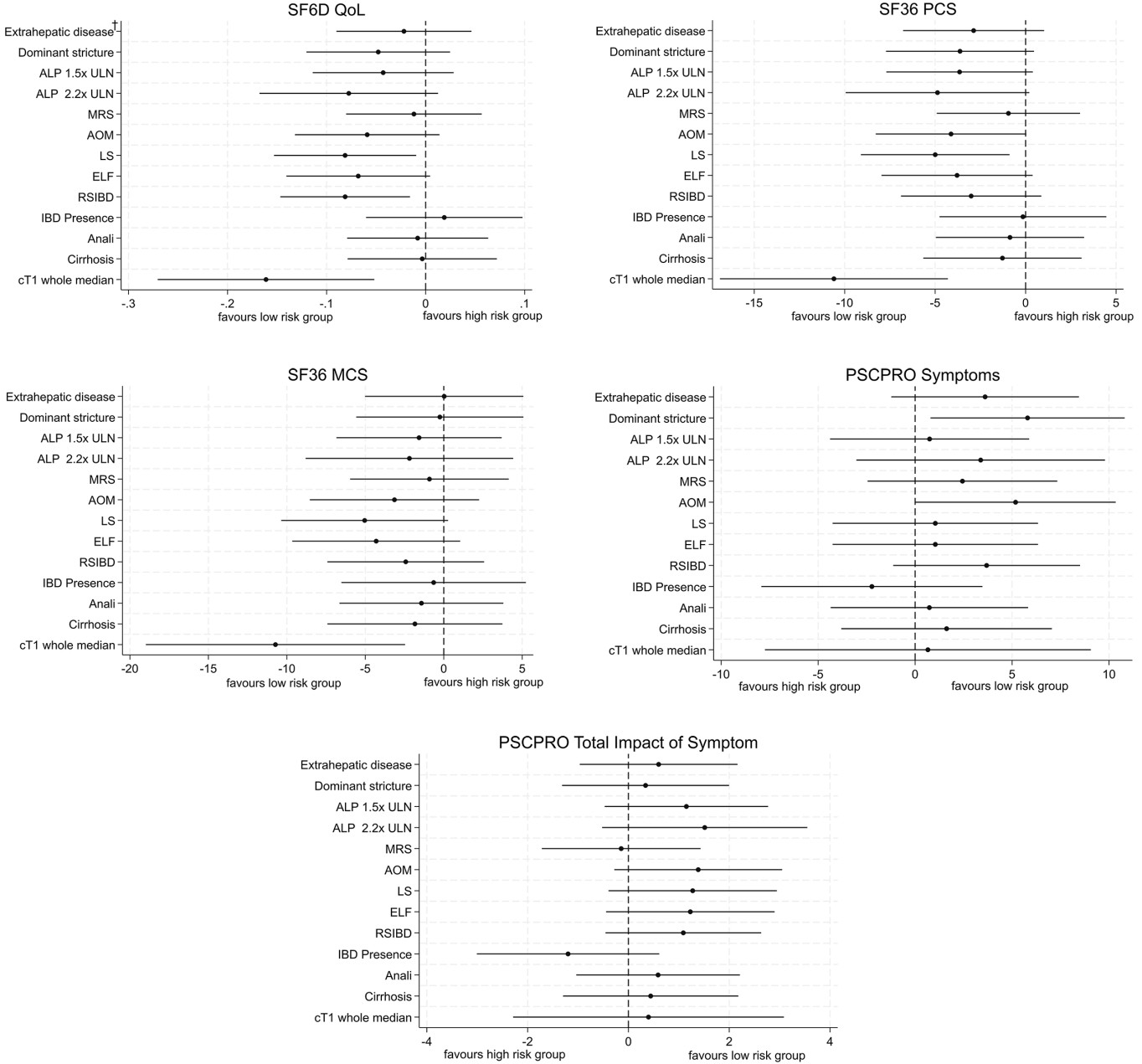

**Fig 2. Differences in PROMs scores at baseline conditional on risk group (n = 75).** † Extrahepatic disease vs. non-Extrahepatic disease; Dominant stricture vs. no Dominant stricture; ULNALP>1.5 vs. ULNALP<= 1.5; ULNALP>2.2 vs. ULNALP<=2.2; MRS > 0 vs. MRS<=0; AOM > 2 vs. AOM<=2; LS > 9.6 vs. LS<=9.6; ELF > 9.8 vs. ELF<=9.8; Relative severity of intrahepatic biliary dilatations; RSIBD>7 vs. RSIBD <=7; IBD presence vs. No IBD; Anali>2 vs. Anali<=2; Cirrhosis vs. no Cirrhosis; cT1 Whole Median > 825ms vs. cT1 Whole Median <= 825ms.

In multivariate analysis, after adjusting for other covariates, changes to Fibroscan LS remained associated with significant changes with SF-36 PCS and SF-6D scores (see Table 4). However, there was no longer a significant association with SF-36 MCS.

There were no significant association between changes in quality of life (SF6D, SF-36 PCS and MCS, and PSC-PRO) and changes in other biomarkers.

In terms of other SF-36 scales' scores, in univariate analysis, we found that an increase in LS was associated with a worsening of the SF36 "Role physical" score (−0.969, p = 0.003). Other biomarkers were significantly associated with some SF-36 scales, such as relative severity of extrahepatic biliary strictures with SF36 Social functioning (−1.161, p = 0.003) and cT1 IQR with SF36 Role physical (0.700, p = 0.003). See S9 Table.

### 3.7 Sensitivity analysis

We found that the likelihood of missing the year 1 visit to be associated with having IBD at baseline (p = 0.016). Following multiple imputation we found our results to be similar to those using available data analysis (See S10 Table for more details). For example, the high risk groups with dominant stricture (5.779, p = 0.019) and AOM > 2 (5.181, p = 0.047), were still associated with higher PSC-PRO symptoms; high risk group LS > 9.6 was associated with lower SF6D QoL (−0.080, p = 0.027) and lower SF36 physical component summary (−4.354, p = 0.036); high risk group with RSIBD>7 was associated with lower SF6D QoL (−0.076, p = 0.020); high risk group with cT1 > 825mm was still find to be associated with lower SF6D QoL (−0.124, p = 0.019) and SF36 physical component summary (−8.243, p = 0.008).

## 4. Discussion

To the best of our knowledge this is the first study to document self-reported quality of life among people with PSC at two time points using a generic instrument and a disease-specific instrument. It is also the first to examine the association between new non-invasive MRI metrics and quality of life.

In this study, as a group, we found a numerical decrease in quality of life after 1 year of follow-up. Our study was small and the follow-up time of 1 year was relatively short for a disease like PSC where the natural history can span decades. However, we found that the presence of dominant stricture, cT1 > 825ms and LS > 9.6kPa to be most impactful in terms of QoL of people living with PSC, both at baseline and year 1 visits. In addition, risk groups based on baseline values of AOM and the MRCP+ derived metric of relative severity of intrahepatic biliary dilations >7 captured variation in QoL at baseline, while extrahepatic disease, ALP (>2.2 ULN) and ELF captured variation in QoL at year 1 visit. However, the risk stratification using baseline biomarkers maybe less clear when accessing changes in QoL between the two visits. Our longitudinal analysis showed that an increase in Fibroscan LS values was associated with a decrease in quality of life, even after adjusting for other covariates. The small sample size and the short follow-up likely explain the lack of statistical significance but also highlight the fact that larger studies with longer follow-up examining changes in QoL are needed.

**Table 4. Association between changes in LS and PROMs.**

| Changes in PROMs scores | Fibroscan LS (increase of 1 kPa), univariate analysis | p-value† | Fibroscan LS (increase of 1 kPa), adjusted multivariate analysis‡ | p-value† |
|---|---|---|---|---|
| SF6D QoL | −0.002 | <0.001*** | −0.002 | <0.001*** |
| SF-36 Physical component summary | −0.246 | <0.001*** | −0.321 | 0.028* |
| SF-36 Mental component summary | −0.171 | <0.001*** | 0.054 | 0.608 |
| PSC-PRO PSC symptoms | 0.179 | 0.275 | 0.503 | 0.399 |
| PSC-PRO total impact of symptoms | 0.063 | 0.057 | 0.096 | 0.151 |

† Hausman test suggest the random effects is more efficient (p > 0.05 for all PROMs scores for univariate and multivariate analysis); the coefficients in table are from random effects model. ‡ *adjusted for* age, MRCP+ number of extrahepatic biliary strictures and dilatations, cT1 IQR, xULNALP, and year. p < 0.05 "*" p < 0.01 "**" p < 0.001 "***".

Previous research in non-alcoholic fatty liver disease (NAFLD), recently renamed metabolic dysfunction associated steatotic liver disease (MASLD) [23] also found LS to be negatively associated with SF-36 scales (Physical function; Role Physical) and EQ-5D-5L utility index [24–26]. These suggest LS as a potentially useful surrogate marker of quality of life across liver disease aetiologies.

Compared with the disease specific PROMs (PSC-PRO), the RAND SF-36 was more sensitive in stratifying QoL over time in people with PSC. For instance, LS changes were found to be associated with changes in the three SF-36 outcomes (SF6D QoL, SF36 PCS, and SF36 MCS) in the analysis. However, LS changes were not associated with changes in the two PSC-PRO scores in the analysis. This could be related with the difference in recall period between the two PROMs, where people with PSC were asked about quality of life in the last 4 weeks in the SF-36 compared to symptoms in the last 24 hours and in the previous week in the PSC-PRO. Studies showed the different time recall period has impact on the PROMs, longer recall period such as 4 weeks may lead people with PSC emphasise more on the salient events, which will get lower overall symptom score and quality of life, but it does not reflect on the variability or disease experience [27–29]. Another possibility is that participants were likely to have postponed their study visits if they had disease-specific symptoms such as cholangitis in the last 24 hours or the preceding 7 days.

MRCP+ generates a wealth of quantitative data relating to the biliary tree. In this study, we only tested one metric that was previously identified as a potential marker of disease severity (relative severity of intrahepatic biliary dilatations; RSIBD, [1]. At baseline, those with more severe disease as defined by this metric (RSIBD > 7) had significantly lower SF6D scores indicating worse quality of life compared with those with RSIBD ≤7. Other traditional MRI measures of disease severity like the presence of a dominant stricture or the presence of extrahepatic disease were associated with QoL suggesting that other biomarkers based on MRCP may be useful correlates of QoL. A more detailed analysis will be needed to evaluate how other MRCP+ metrics relate to the quality of life. People with PSC and liver cT1 > 825ms were also found to have worse quality of life and therefore other quantitative MR markers would be worth of future evaluation.

A systematic review [8] identified previous work in PSC to consist of cross-sectional studies, where there was no consistent association between QoL in people with PSC and PSC surrogate markers of disease severity (ALP, cirrhosis, MELD). However, in our study, we benefitted from having biomarkers captured at two time points. This allowed identification of the inverse relationship between Fibroscan liver stiffness and QoL even after the short follow-up period of 1-year.

Our study is not without limitations. Not all participants that attended the baseline visit were able to attend the year 1 visit due to COVID-19 lockdown. Hence, the PROMs were missing for 24 participants. However, in our sensitivity analysis, after we did a multiple imputation to fill in the missing data, we find that imputed data set analysis shows the similar results compare with the available dataset. Furthermore, there was a numerical difference of 10% in the proportion of participants with IBD at baseline and 1 year. While this did not reach statistical significance due to small numbers in our study, it may become significant when larger studies are considered. Another limitation was having participants enrolled from a single study site and the sample being relatively small. The small sample size reflects the rare nature of the disease and constrained our statistical analysis in terms of power. Hence, to avoid reporting results that could not be transferred outside the sample, we adopted a conservative approach to deciding on the significance of the results using a two-step approach (bootstrap and Cook's distance). Larger samples in PSC populations would be useful to confirm our findings.

In summary, PSC is a rare chronic liver disease that can affect quality of life and our study has potential important clinical implications. Knowing whether biomarkers are associated to disease severity as well as QoL can inform clinicians and health system how to use biomarkers in clinical care. By choosing to use tools that can inform both for the disease severity and the impact this is having on the quality of life care can become more efficient. For example, by employing a tool that can measure bith disease severity and QoL can negate the need for using separate tools for for monitoring disease severity and questionnaires to measure QoL,. We found several markers including RSIBD measured by MRCP+

and liver cT1 measured by Liver MultiScan to be cross sectionally associated with QoL. In addition, changes in LS were also associated with changes in QoL in people with PSC. Further work should evaluate how the wealth of quantitative MRI data relate to quality of life to further inform the potential role of these parameters in the management of people with PSC. PROMs are important beyond clinical and imaging findings and provide key insight into people with PSC' perspectives and lived experiences.

## Supporting information

**S1 Table. Type of data collected.**
(PDF)

**S1 Text. Bootstrap approach to assess robustness of findings.**
(PDF)

**S2 Text. Cook's distance to identify potential outliers.**
(PDF)

**S3 Text. Multiple imputation for sensitivity analysis.**
(PDF)

**S2 Table. Imputation model.**
(PDF)

**S1 Fig. Study flowchart.**
(PDF)

**S3 Table. Descriptive statistics and comparison between different sample sizes.**
(PDF)

**S4 Table. PROMs comparison between different sample sizes.**
(PDF)

**S5 Table. The within difference of patients' PROMs.**
(PDF)

**S6 Table. Baseline PROM scores by risk group.**
(PDF)

**S7 Table. PROM scores at year 1 by baseline PSC disease severity group.**
(PDF)

**S8 Table. Change in PROM scores by baseline PSC severity.**
(PDF)

**S4 Text. Robustness check results.**
(PDF)

**S9 Table. Univariate analysis for other PROMs' domains.**
(PDF)

**S10 Table. Sensitivity analysis (Multiple imputation).**
(PDF)

## Author contributions

**Conceptualization:** Emmanuel Selvaraj, Michael Pavlides, Jose Leal.

**Data curation:** Emmanuel Selvaraj, Michael Pavlides.

**Formal analysis:** Jingyu Dai.

**Funding acquisition:** Michael Pavlides, Jose Leal.

**Investigation:** Jingyu Dai, Michael Pavlides, Jose Leal.

**Methodology:** Jingyu Dai, Jose Leal.

**Project administration:** Jose Leal.

**Software:** Jingyu Dai.

**Supervision:** Jose Leal.

**Validation:** Jingyu Dai.

**Visualization:** Jingyu Dai.

**Writing – original draft:** Jingyu Dai.

**Writing – review & editing:** Emmanuel Selvaraj, Emma L Culver, Adam Bailey, Michael Brady, Michael Pavlides, Jose Leal.

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
