## [Decision Letter · Decision Letter 0]

24 Jul 2025

Dear Dr. Leal,

Thank you for submitting your manuscript to PLOS ONE. After careful consideration, we feel that it has merit but does not fully meet PLOS ONE’s publication criteria as it currently stands. Therefore, we invite you to submit a revised version of the manuscript that addresses the points raised during the review process.

We look forward to receiving your revised manuscript.

Kind regards,

Olivier Barbier

Academic Editor

PLOS ONE

Journal Requirements:

“JD, JL, ELC, MP and AB supported by Oxford NIHR Biomedical Research Centre. (https://oxfordbrc.nihr.ac.uk/)

ELC receives funding from PSC Support and Oxford Health Service Research Committee.”

3. Please expand the acronym “NIHR” (as indicated in your financial disclosure) so that it states the name of your funders in full.

“This study is funded by Oxford NIHR BRC imaging theme.”

“JD, JL, ELC, MP and AB supported by Oxford NIHR Biomedical Research Centre. (https://oxfordbrc.nihr.ac.uk/)

ELC receives funding from PSC Support and Oxford Health Service Research Committee.”

“I have read the journal's policy and the authors of this manuscript have the following competing interests: MB is employed by Perspectum Ltd and is a shareholder in Perspectum Ltd, MP is a shareholder in Perspectum Ltd.

ELC: Speaking Fees: Advanz (Intercept), Albireo, Dr Falk Pharma, Gilead, GSK, Mirum; Consulting Fees: Advanz (Intercept), Amgen (Horizon Therapeutics), Ipsen, Mirum, Moderna, Sanofi, Zenus Pharma, Sail; Grant Support: Jansen, Innovate UK, PSC Support; Institutional Funding Support: BRC Oxford NIHR (UK), Oxford Health Service Research Committee Grant.”

We note that you received funding from a commercial source: Perspectum Ltd

6. Please include a separate caption for each figure in your manuscript.

Reviewers' comments:

Reviewer's Responses to Questions

**Comments to the Author**

1. Is the manuscript technically sound, and do the data support the conclusions?

Reviewer #1: Yes

Reviewer #2: Yes

2. Has the statistical analysis been performed appropriately and rigorously?

Reviewer #1: Yes

Reviewer #2: Yes

3. Have the authors made all data underlying the findings in their manuscript fully available?

Reviewer #1: Yes

Reviewer #2: Yes

4. Is the manuscript presented in an intelligible fashion and written in standard English?

Reviewer #1: Yes

Reviewer #2: Yes

Reviewer #1: Your study was well done and will be of interest to a spectrum of care givers, basic scientists, epidemiologists, pharmaceutical companies and governmental agencies. I have nothing further to add. .

Reviewer #2: The authors conducted a study with 80 PSC patients total and examined whether sveral biomarkers, MRCP+ metrics and risk scores correlate with changes in QoL. To my knowdledge there is no other study which analyzed this combination of markers. The authors collected data at basline and after 1 year. They have identified markers which are connected to a more advanced disease and the identified that an increase in liver stiffness correlates with reduced QoL indicators.

The limitations of this study are the sample size and the fact that they only received data at 2 timepoints and only with a one year gap.Since PSC is a slow progressing disease, one year is a short time. But, in the discussion the authors adressed the same concerns.

Otherwise the technical aspects and stastics are sound and reasonable, although the authors could eloborate more why they chose exactly those regression models. Moreover, they used standard QoL measurements and biomarkers/risk scores which are well documented. Whats new is the combinations of markers and the correlation to QoL.

From my perspective, a more eloborate statement regarding the impact of the study on clincal practice is missing. Why and how is it useful to now which markers are connected to a reduced QoL? Additionally, an outlook would be interesting, so whether the authors plan to repeat the study with more patients at longer intervals. Since they indicated that sampling is ongoing.

Points to revise:

- Table 1: ¶ this symbol was used twice for different markers, which is a bit confusing

- 2.3.1: Module 1 consists of 13 question with 10 being the highest score. Should it not add up to 130 and not 120?

- 2.3.2: the information is in the manual, but shortly mention which scales were combined for easier understanding

- 3.1: while the IBD prescence is not significant between the basline and 1 year, authors should mention that 10 % is still

a modest difference and might be significant when a larger sample size is used

**Do you want your identity to be public for this peer review?** For information about this choice, including consent withdrawal, please see our Privacy Policy

Reviewer #1: No

Reviewer #2: No

---

## [Author Response · Author response to Decision Letter 1]

30 Sep 2025

Journal Requirements:

AUTHORS’ REPLY: Thank you for the comment. We changed the manuscript to meet the style requirements, including changes of files name.

“JD, JL, ELC, MP and AB supported by Oxford NIHR Biomedical Research Centre. (https://oxfordbrc.nihr.ac.uk/)

ELC receives funding from PSC Support and Oxford Health Service Research Committee.”

AUTHORS’ REPLY: Thank you for the comment. We have now added the funders’ role into the Funding statement: "The funders had no role in study design, data collection and analysis, decision to publish, or preparation of the manuscript."

We also add one sentence in the cover letter: “We confirm that the funders had no role in study design, data collection and analysis, decision to publish, or preparation of the manuscript.”

3. Please expand the acronym “NIHR” (as indicated in your financial disclosure) so that it states the name of your funders in full.

AUTHORS’ REPLY: Thank you for the comment. NIHR stands for National Institute for Health and Care Research which is now expanded in the paper. We have also added the funder’s information to the cover letter.

“This study is funded by Oxford NIHR BRC imaging theme.”

“JD, JL, ELC, MP and AB supported by Oxford NIHR Biomedical Research Centre. (https://oxfordbrc.nihr.ac.uk/)

ELC receives funding from PSC Support and Oxford Health Service Research Committee.”

AUTHORS’ REPLY: We have removed the funding information as requested from the manuscript (Funding statement and Conflict of interest disclosure) and added it to the cover letter.

“I have read the journal's policy and the authors of this manuscript have the following competing interests: MB is employed by Perspectum Ltd and is a shareholder in Perspectum Ltd, MP is a shareholder in Perspectum Ltd.

ELC: Speaking Fees: Advanz (Intercept), Albireo, Dr Falk Pharma, Gilead, GSK, Mirum; Consulting Fees: Advanz (Intercept), Amgen (Horizon Therapeutics), Ipsen, Mirum, Moderna, Sanofi, Zenus Pharma, Sail; Grant Support: Jansen, Innovate UK, PSC Support; Institutional Funding Support: BRC Oxford NIHR (UK), Oxford Health Service Research Committee Grant.”

We note that you received funding from a commercial source: Perspectum Ltd

AUTHORS’ REPLY: Thank you for this comment and we apologise if this was not clear in the initially amended manuscript. We did not receive any funding from Perspectum. Perspectum provided analysis for MRCP+ and liver multiscan data as contribution in kind. We have added this to our conflict-of-interest statement in Cover letter.

6. Please include a separate caption for each figure in your manuscript.

AUTHORS’ REPLY: We have now included separate captions in the manuscript.

The captions of my supporting information are now at the end of my manuscript, and in the format the Journal asked.

AUTHORS’ REPLY: We have reviewed the reference list.

Reviewers' comments:

Reviewer's Responses to Questions

9. (09/09/2025) In this instance it seems there may be acceptable restrictions in place that prevent the public sharing of your minimal data. However, in line with our goal of ensuring long-term data availability to all interested researchers, PLOS’ Data Policy states that authors cannot be the sole named individuals responsible for ensuring data access (http://journals.plos.org/plosone/s/data-availability#loc-acceptable-data-sharing-methods).

AUTHORS’ REPLY: We thank the Editor for the helpful comments regarding data availability. The dataset used for all analyses is now publicly available on the Open Science Framework (OSF) (DOI: https://doi.org/10.17605/OSF.IO/BH94N). The Data Availability Statement and CL have been revised to reflect this update.

Comments to the Author

1. Is the manuscript technically sound, and do the data support the conclusions?

Reviewer #1: Yes

Reviewer #2: Yes

2. Has the statistical analysis been performed appropriately and rigorously?

Reviewer #1: Yes

Reviewer #2: Yes

3. Have the authors made all data underlying the findings in their manuscript fully available?

Reviewer #1: Yes

Reviewer #2: Yes

4. Is the manuscript presented in an intelligible fashion and written in standard English?

Reviewer #1: Yes

Reviewer #2: Yes

5. Review Comments to the Author

Reviewer #1: Your study was well done and will be of interest to a spectrum of care givers, basic scientists, epidemiologists, pharmaceutical companies and governmental agencies. I have nothing further to add. .

Reviewer #2: The authors conducted a study with 80 PSC patients total and examined whether sveral biomarkers, MRCP+ metrics and risk scores correlate with changes in QoL. To my knowdledge there is no other study which analyzed this combination of markers. The authors collected data at basline and after 1 year. They have identified markers which are connected to a more advanced disease and the identified that an increase in liver stiffness correlates with reduced QoL indicators.

The limitations of this study are the sample size and the fact that they only received data at 2 timepoints and only with a one year gap.Since PSC is a slow progressing disease, one year is a short time. But, in the discussion the authors adressed the same concerns.

Otherwise the technical aspects and stastics are sound and reasonable, although the authors could eloborate more why they chose exactly those regression models. Moreover, they used standard QoL measurements and biomarkers/risk scores which are well documented. Whats new is the combinations of markers and the correlation to QoL.

From my perspective, a more eloborate statement regarding the impact of the study on clincal practice is missing. Why and how is it useful to now which markers are connected to a reduced QoL? Additionally, an outlook would be interesting, so whether the authors plan to repeat the study with more patients at longer intervals. Since they indicated that sampling is ongoing.

AUTHORS’ REPLY: We thank the reviewer for this important comment. We have revised our closing paragraph to emphasise the potential clinical implications of our work. We already provide suggestions for future work and we will update our results once our study is completed. At this point in time, we have not secured funding for a study extension beyond the 1-year follow up.

Our last paragraph (Page 22) now reads:

“In summary, PSC is a rare chronic liver disease that can affect quality of life and our study has potential important clinical implications. Knowing whether biomarkers are associated to disease severity as well as QoL can inform clinicians and health system how to use biomarkers in clinical care. By choosing to use tools that can inform both for the disease severity and the impact this is having on the quality of life care can become more efficient. For example, by employing a tool that can measure bith disease severity and QoL can negate the need for using separate tools for monitoring disease severity and questionnaires to measure QoL. Surrogate biomarkers may be helpful in stratifying people with PSC according to their quality of life besides the risk of disease progression. We found several markers including RSIBD measured by MRCP+ and liver cT1 measured by Liver MultiScan to be cross sectionally associated with QoL. In addition, changes in LS were also associated with changes in QoL in people with PSC. Further work should evaluate how the wealth of quantitative MRI data relate to quality of life to further inform the potential role of these parameters in the management of people with PSC. PROMs are important beyond clinical and imaging findings and provide key insight into people with PSC’ perspectives and lived experiences.”

Points to revise:

- Table 1: ¶ this symbol was used twice for different markers, which is a bit confusing

AUTHORS’ REPLY: Thank you for the comment, in Table 1, ¶ symbol is now only used for ULN.

- 2.3.1: Module 1 consists of 13 question with 10 being the highest score. Should it not add up to 130 and not 120?

AUTHORS’ REPLY: The question: “Are you currently experiencing a flare-up of your PSC symptoms (also known as acute cholangitis or an infection in the bile ducts)? Y/N”, does not produce a score.

- 2.3.2: the information is in the manual, but shortly mention which scales were combined for easier understanding

AUTHORS’ REPLY: We have added the following sentence (page 9):

“Physical component summary (PCS) and mental component summary (MCS) scores are aggregated by applying a scoring algorithm combines weighted contributions from all eight health scales.”

- 3.1: while the IBD prescence is not significant between the baseline and 1 year, authors should mention that 10 % is still a modest difference and might be significant when a larger sample size is used

AUTHORS’ REPLY: We agree with the reviewer and have added the following to our discussion (page 22):

“Furthermore, there was a numerical difference of 10% in the proportion of participants with IBD at baseline and 1 year. While this did not reach statistical significance due to small numbers in our study, it may become significant when larger studies are considered.”

6. PLOS authors have the option to publish the peer review history of their article (what does this mean?). If published, this will include your full peer review and any attached files.

Do you want your identity to be public for this peer review? For information about this choice, including consent withdrawal, please see our Privacy Policy.

Reviewer #1: No

Reviewer #2: No

---

## [Editor Report · Decision Letter 1]

15 Oct 2025

Association between non-invasive biomarkers and quality of life in primary sclerosing cholangitis

PONE-D-25-19576R1

Dear Dr.*Jose Leal*

We’re pleased to inform you that your manuscript has been judged scientifically suitable for publication and will be formally accepted for publication once it meets all outstanding technical requirements.

Kind regards,

Olivier Barbier

Academic Editor

PLOS ONE

---

## [Editor Report · Acceptance letter]

PONE-D-25-19576R1

PLOS ONE

Dear Dr. Leal,

I'm pleased to inform you that your manuscript has been deemed suitable for publication in PLOS ONE. Congratulations! Your manuscript is now being handed over to our production team.

Kind regards,

on behalf of

Prof. Olivier Barbier

Academic Editor

PLOS ONE